# Polymorphic Biological and Inorganic Functional Nanomaterials

**DOI:** 10.3390/ma15155355

**Published:** 2022-08-03

**Authors:** Tessa Gilmore, Pelagia-Irene Gouma

**Affiliations:** Department of Material Science and Engineering, The Ohio State University, Columbus, OH 43210, USA; gouma.2@osu.edu

**Keywords:** polymorphism, nanomaterials, amyloids, electrospinning

## Abstract

This perspective involves two types of functional nanomaterials, amyloid fibrils and metal oxide nanowires and nanogrids. Both the protein and the inorganic nanomaterials rely on their polymorphism to exhibit diverse properties that are important to sensing and catalysis. Several examples of novel functionalities are provided from biomarker sensing and filtration applications to smart scaffolds for energy and sustainability applications.

## 1. Introduction

There is recent interest in studying biologically mediated self-assembly to create novel configurations of inorganic materials with organic components, which allows for the production of complex, hierarchical, 3D architectures with 4D functionality. We feature here examples of nanowire sensors targeting gaseous biomarkers that may lead to skin sensing of metabolites for automatic, non-invasive, non-intrusive health monitoring. These sensors will be especially useful for the early detection of exposure to biochemical agents and other pathogens. Furthermore, novel, nanostructured, self-supported, visible-light-activated photocatalysts will be developed for water remediation and for water splitting to create hydrogen and oxygen fuel, while also capturing free CO_2_. Mechanical, optical, and chemical stimuli will fine tune and/or reverse the functionality of the new material systems. Biomedical uses of the novel constructs include nanoprobes targeting the headspace above cells for signaling biomarkers of disease (volatolomics); theragnostic tools and therapeutic formulations, among others.

## 2. Functional Protein Nanomaterials: Amyloids

Amyloids have emerged as important functional biomaterials, as their structures can be tailored to accommodate a wide range of inorganic materials via doping and functionalization. As a result, they can act as bio-voxels for the bottom-up assembly of novel and unique material architectures and properties related to structural, optical, electronic, and catalytic effects [1]. The challenge has been to achieve controlled and guided assembly of the amyloid fibrils into the complex, 3D architectures that are required for the unique functionality of the amyloid-modified materials [2]. In the literature, amyloid polymorphism has been identified [3] as a key feature controlling the relative affinity of the fibril surfaces.

Polymorphism results from the different relative arrangement of identical sequences of proteins or peptides as they self-assemble into fibrils. Polymorphism can arise simply by varying the arrangements of β-strands within a β-sheet. Due to its importance, it is desirable to map the distinct self-assembled structures (i.e., establish the design rules). Therefore, understanding and controlling amyloid polymorphism to produce predictable and reproducible phases is a key task for any future work in this field. This polymorphism allows for binding to various organic and inorganic species and the ability of their structural components (-sheet and interchain structures) to form intersheet cross-links with functionalized chemical groups. The main challenge to address is the guided amyloid self-assembly at the macroscale in the desired patterns.

The amyloid fibrils we have synthesized have diameters in the 10 nm range and lengths less than 100 nm [4]. Therefore, they are the ideal templates (bio-voxels) for enabling novel material phenomena, which affect mechanical strength and the photoactivated and pressure-activated properties of materials. Several processes that allow the formation of aligned nanofibers of high aspect ratio and the formation of 3D hierarchical designs are highlighted in this perspective, including the electrostatic drawing of fibrous mats from hybrid hydrogels. The inorganic materials to emerge from amyloids in the specific assemblies can be dynamically and reversibly modified. There is a need to produce fundamental knowledge on the following: biological-inorganic material interactions, novel 4D manufacturing based on bio-voxel printing, and superior catalytic, electronic, optical, and biological functionalities.

Amyloid fibrils were first observed by the 19th century’s prominent German physician Rudolph Virchow in 1854 [5]. Pauling and Corey [6] proposed what is now the prevalent model of amyloid fibril structure, consisting of the cross-β core, with β-strands embedded in β-sheets and parallel to the fiber axis, where β-sheets lie perpendicular to the major axis of the fiber [7]. Amyloid fibrils can be obtained by the stable unfolding of functionally folded peptides, as well as proteins [7]. Figure 1 represents the various stages involved in amyloid fibril formation under suitable physico-chemical conditions. Amyloid fibril formation occurs from intermediate unfolded peptides and protein structures into a stacking of β-strands [7].

### 2.1. Amyloid Fibrils: From Inducing Disease to Becoming a Biomaterial with Diverse Applications

Recent studies revealed that pre-fibrillar amyloid assemblies might contain toxic elements, which are responsible for cell poisoning [9]. Diseases such as Alzheimer’s disease, Huntington’s and Parkinson’s disease occur due to the accumulation of amyloid fibrils in tissues [10]. As amyloid fibrils are insoluble and resistant to degradation, they are implicated with causing the diseases mentioned above [11,12]. However, analysis of the amino acid sequence and composition revealed that only particular proteins and peptides are responsible for amyloid disorders [13]. Furthermore, non-pathogenic functional amyloid aggregates control important biological processes in diverse organisms, from bacteria to humans. The physical properties of amyloid fibrils mean that they have strength similar to steel and silk fibers [14]. Moreover, they have a high Young’s modulus, attributed to the dense network of hydrogen bonding, which, among other things, leads to immense interaction between the backbones of polypeptides [15,16,17].

Studies focused on the behavior of amyloidogenic motif sequences provide better insight of their self-assembly mechanism [18,19]. These studies are providing the basis for designing bio-modified nanomaterials based on amyloids. Wetzel et al. [20] summarized the crucial similarities between amyloids, synthetic polymers and plastics to be the following:(1)The assembly properties of amyloid and its polymer subunits do not change under the influence of major chemical modifications;(2)comparable isomorphism can be obtained from various monomeric units;(3)condensed state is formed via noncovalent interaction (…); and(4)Under specific conditions gel or liquid crystals can form.

Thus, the unique and versatile properties of amyloid fibrils provide ample opportunities for technological development and for making nanomaterials via “bottom-up” approaches. Special features, such as higher structural stability, nanoscale dimensions, and ease of production, make amyloid fibrils a suitable nano-biomaterial template for inorganic crystal growth. The amyloid fibril can drive the advances in nanobiotechnology and associated applications because of its unique properties as protein nanomaterial.

### 2.2. Recent Breakthrough: Novel Process for Amyloid from Wheat Flour

Wheat flour contains different proteins that can be classified as albumins, globulins, gliadins, and glutenins [21]. Using Osborne solubility rules [22], each group of proteins was separated out. SDS-insoluble subunits of glutenins were the only proteins to form amyloid fibrils under acidic conditions. Therefore, the aim of the separations was to isolate those specific subunits [4]. Thioflavin T (ThT) is a benzothiozole dye with an increased inclination towards β-sheets containing proteins and is widely considered the standard for identifying fibers in ex vivo, in vitro, and in animal model studies [23]. Unbound ThT dye has fluorescence excitation from 385 nm to 450 nm. Upon binding to a β-sheet structure, the dye experiences a characteristic spectral shift, which results in heightened fluorescence emission from 445 nm to 482 nm. This change in the spectral shift is employed for the bifurcation of bound and unbound ThT, as well as the presence of amyloid fibrils’ structure [24]. In the confocal images, amyloids appear green; however, in the TEM images, the amyloid fibrils appear dark. This difference is due to the negative staining obtained from uranyl acetate. Figure 2 shows a confocal image and a TEM micrograph, respectively. It is evident that uniformly dispersed amyloid fibrils are present, with diameters ranging from 7 to 10 nm and lengths of 100 nm. Therefore, the materials obtained by the novel synthesis method are in the 10–100 nm range, the critical range for realizing novel material phenomena.

### 2.3. Amyloid Composites with Super-Hydrophobic CA Electrospun Mats

In our recent research [25], super water repellant cellulose acetate (CA) fibrous mats were developed using electrospinning in a one step process, without further surface modification. The as-spun CA fibrous mats show low density, high surface roughness, and a high surface-to-volume ratio (Figure 3). The effect of electrospinning on the roughness of the surface was confirmed through applying the Cassie’s model on the as-spun fibers. It is shown that the roughness effect alone in this process cannot induce the shift from hydrophilic to super water repellant. Furthermore, it can be observed that the intensity of the hydroxyl band at 3450 cm^−1^ is decreased in the as-spun fiber, meaning that it may be possible that the concentration of the hydroxyl groups on the electrospun fiber is decreased during the electrospinning process.

Heavily charged or hydrophobic surfaces can catalyze the formation of amyloid aggregates [26]. Amyloid-like structures are usually prepared in-vitro by processing insulin (by heating it in acidic pH). The so-processed amyloid-like fibrous structures were blend-electrospun with cellulose acetate solutions (see recipe in Figure 4 below) and the resulting nanofibrous configurations were tested for water remediation properties.

The processing steps for the amyloid fibrils embedded in cellulose acetate (CA) mats are as follows:Materials: cellulose acetate (MW 30,000), amyloid fibrils (prepared from bovine insulin), acetic acid, acetone;A total of 3 mL of 15 wt.% CA solution was mixed with 1 mL of amyloid fibril-containing solution;Electrospinning: 1 mL/hr flow rate, 7 cm working distance, 20 kV voltage.

## 3. Functional Inorganic Nanomaterials

### 3.1. Nanowires of α-MoO_3_

Novel synthesis of continuous single crystal nanowires of α-MoO_3_ was demonstrated in our earlier work using a single step process [27,28]. It involved modifying the sols of metal oxides through their interactions with a carrier polymer, upon being released from a metallic orifice under the force of an electrostatic field. The process, known as electrospinning, operates by overcoming the surface tension of the liquid droplet of a solution mixture to create a continuous jet, while the solvent evaporates in flight. This leaves on the metallic collector solid continuous fibers of a core-shell morphology, in a single step without using any special orifice; the core being the amorphous metal oxide. During calcination, and as the polymer is decomposing, a massive-type phase transformation converts the amorphous core to a continuous, single crystal of nanoscale diameter and micro-scale length [28]. Their dimensions are 10–15 nm in width and more than 2 um long. The nanowires obtained by this process (see Figure 5) have a high aspect ratio and defect free microstructures. The measured d-spacings for the nanowires shown in these figures were 6.944 Å, 3.9 Å, and 1.822 Å, which correspond to the (020), (100), and (230) planes of the orthorhombic α-MoO_3_ polymorph, respectively. The crystal belongs to the space-group Pbnm [29]. These novel nanoarchitectures promote the material’s selectivity and sensitivity to ammonia gas sensing.

To assess the sensing response of the MoO_3_ nanowire mats to NH_3_ and compare them with that of sol-gel-based thin films of nanopowders stabilized under the same conditions, calcined (20/80/0.5 M) MoO_3_/PVP electrospun mats were ultrasonically agitated in ethanol for 5 min before sensing. The sensors were then placed in a quartz tube inside of a tube furnace (Lindberg/Blue) and heated to 450 °C at a programmed rate. The sensors were allowed to stabilize in gaseous mixtures of UHP oxygen (Praxair Inc., Danbury, CT, USA) and UHP nitrogen (Praxair) with an oxygen concentration of 20% in the inlet stream. The analyte of interest was chosen to be NH_3_, as previous studies had observed the α-orthorhombic phase of MoO_3_ selectively detecting NH_3_ within interfering gaseous compounds, such as NO, CO, etc. The resistance in both the nanowire and the sol-gel sensors decreased, which confirmed that both types of sensors exhibit n-type semiconductor behavior.

The results from the sensing tests reveal that the sensitivity of the nanowire mat increases with increasing the length of the nanowires [28]. There was an order of magnitude improvement in gas detection sensitivity compared to sol-gel processed powder materials of the same diameter. While sol-gel-based sensors had a detection threshold of 50 ppb for ammonia gas, the nanowire-based equivalent could detect concentrations down to a few ppbs, which is more than sufficient to detect ammonia emitted from the body (breath, skin) [28].

### 3.2. The 4D Functionality of Bio-Nano-Materials-Volatolomics: Headspace Chemo-Sensing

Sensing of volatile organic compounds (VOCs) emitted by cells in response to chemical and mechanical perturbations can open the pathway for the rapid detection of infectious and chronic diseases by sampling the skin or breath of humans and animals. Novel sensing probes for measuring VOCs emitted from cell lines and electronic nose-based sensor systems consisting of electrospun oxide nanowire sensing elements grown on amyloid fibrils are foreseen for the near future. Augmenting our understanding of continuous single crystal nanowire formation guided by organic materials, as presented above (see patent [30]), is necessary. Utilizing hybrid gels of amyloids with MoO_3_ sols, for example, will produce single crystals of MoO_3_ with well-defined widths and lengths and tailored polymorphisms. The aim is to obtain uniform oxide nanowires, with a tailored crystal structure to control gas selectivity, giving an amplified sensing response that will allow us to detect trace concentrations of various VOC analytes of interest with high specificity.

The specific nano-assemblies envisioned will also enable other functional applications of α-MoO_3_, such as its use as a cathode in Mg batteries. Layered MoO_3_ offers one of the highest theoretical capacities for Mg^+2^ intercalation; however, thin films of this material suffer from slow diffusion kinetics [31]. The self-assembly of single crystal nanowires of α-MoO_3_ is expected to provide “clean” intercalation paths for the fast and efficient reversibility of Mg^+2^ [31]. Furthermore, integration of amyloid-modified electronic sensing probes into constructs (fabrics; tools; devices) for non-invasive skin monitoring of health; rapid detection of live viruses in the air; as well as in 3D nanostructured, self-supported, energy producing, converting, and storage systems, will be possible.

### 3.3. Hydrothermal Synthesis of Metastable Oxide Phases-Nanowires of Hexagonal-WO_3_

While via electrospinning only the thermodynamically stable phases of oxides are obtained, the diverse polymorphism of amyloids will be explored to facilitate the growth of metastable oxide polymorphs, such as the perovskite-structured β-MoO_3_ and the hexagonal form of the same oxide, via hydrothermal processing [32]. The latter was used in our recent studies to produce semiconducting nanowires of the h-WO_3_ for the detection of VOCs, such as isoprene and acetone. Figure 6 provides a process map for the processing of the metastable polymorphs of metal oxides. The simultaneous hydrothermal treatment of amyloid fibers and h-WO_3_ nanowires should be explored, as environmental factors, such as the solution pH, ionicity, etc., also affect amyloid polymorphism.

### 3.4. Electron Lithography for Nanowire Growth—The Case of γ-WO_3_


Novel processing of γ-WO_3_ nanowires from metastable precursors under electron irradiation has produced polytypic nanowires in a rapid process (Figure 7). The single-phase, one-dimensional structures show polytypism as manifested in the high-resolution TEM and the selected area diffraction pattern [33]. A soft lithography technique was demonstrated by other workers to pattern amorphous “generic amyloid inks”, consisting of CsGA proteins into well-defined arrays of β-sheet structures post-curing [5]. Amyloids have the added advantage over DNA-type bio-voxels in that they can withstand harsh environments and high temperatures, thus are amenable to a number of fabrication routes [5]. Self-supported and bio-inspired patterns may be explored for amyloid lithography. Growing oxide nanowires in situ using electron lithography on patterned amyloid templates or from amyloid-hybrid hydrogels is expected to provide one-of-a-kind homogeneity in dimensions, size distribution, fine-tuned crystallinity, and control of the stacking fault formation, which causes the observed polytypism. Novel material configurations will be obtained for sensing and catalytic applications of high efficiency.

## 4. Smart and Hybrid Scaffolds

Electrospun foams are illustrated in Figure 8 below [34]. Although the literature stated that these foams were based on a honeycomb structure, the structure is more closely related to a 3D fiber-woven foam. Self-supported tungsten oxide (WO_3_) foams were synthesized by a combination of sol-gel, electrospinning, and thermal oxidation processes. Mixtures of tungsten isopropoxide (C_18_H_42_O_6_W)-based precursors and cellulose acetate (CA) were electrospun and subsequently heat-treated. Structural characterization of the as-processed foam-like monoliths confirmed that they consist of cubic WO_3_ nanoparticles in a continuous matrix with open porosity. The formation of the self-supported nano-foams is a result of self-assembly of the composite nanofibers in non-woven electrospun mats. The cubic WO_3_ foams have a band-gap of 2.53 eV and they demonstrated catalytic action when activated by visible-light. Adding catalytic properties to a 3D scaffold that are externally stimulated is a novel property that needs to be further explored. Furthermore, exploring piezoelectric oxide materials that generate and transfer bioelectric signals, in a similar way to native tissue, when tensile/compression forces act on them is very desirable, but is yet not possible with the current manufacturing processes [35]. Such smart scaffolds with catalytic and bio-electric functionality are highly desirable. The novel techniques highlighted here for material processing can produce such structures in a single step.

### 4.1. The 3D Scaffolds of Self-Supported Photocatalysts for Water Clean Up and CO_2_ Capture

The advantage of using semiconductor-based photocatalysts to remediate contaminated water is that complete mineralization of certain pollutants is possible. The photocatalytic oxidation reaction (see Equation (1) [36]) allows for petroleum hydrocarbon to be converted to less harmful components, such as carbon dioxide, water, and water-soluble organics that are biodegradable by marine bacteria.
RCH_2_CH_2_ R’ + ˙OH → RĊHCH_2_ R’ + H_2_O(1)

Previous work has been completed in creating a directed self-assembly process consisting of metal diffusion inside of electrospun nanofiberous mats [37], the result of which formed 3D macroscale mats. CuWO_4_ is a metal oxide photocatalyst that utilizes a longer light wavelength (band gap: 2.3 eV) with high photostability in neutral pH. CuWO_4_ uses the OH radicals from electron-hole separation from the oxidation of water and hydroxyl ions. These oxidizing species, as well as other reactive species, such as H_2_O_2_, are able to degrade the polluting substance. The best response for decomposing benzene in water under visible light was found when using CuWO_4_ with CuO [38]. Other researchers [39] also inferred that tungstates of Cu change the valance band with 3D orbitals, which helps with absorption in visible light.

### 4.2. The 3D Fabrication of Nanogrids 

In order to create the sols for the solution, water was added to 1.5 g of tungsten isopropoxide (C_18_H_42_O_6_W) [38]. Hydrolysis was completed inside of a glovebox, where the resulting solution was then mechanically agitated for 5 min. The solutions underwent ultrasonication for 2 h before aging for 24 h to allow for complete hydrolysis. Afterwards, 1.5 g of WO_3_ sol-gel was combined with 3 mL of acetic acid and 3 mL of ethanol within a glovebox filled with nitrogen. The mixture was then removed from the glovebox and added to a solution of 10% wt/vol polyvinylpyrrolidone PVP (Aldrich, MW~1,300,000) in ethanol, and then placed in an ultrasonic bath for around 30 min.

Once completed, 5 mL of the mixture was immediately placed into a syringe positioned vertically 7 cm above a piece of copper mesh (TWP Inc., Berkeley, CA, USA, 200 mesh, wire dia. 51 μm). The syringe needle was connected to a high voltage power supply and the copper mesh acted as a ground electrode. The syringe pump was programmed for a flow rate of 30 μL/min. Once a high voltage was applied to the syringe needle (25 kV), a solution jet was formed at the needle tip and the solvent evaporated in flight, resulting in a nonwoven mat of fibers deposited on the Cu mesh.

After the electrospinning process had finished, complete calcination of the PVP nanofibers via thermal oxidation was performed at 500 °C for 5 h. This oxidation process occurs by first driving CuO crystals into the PVP nanofibers, which already encompass amorphous WO_3_. As the process evolves, crystals of WO_3_ form between and around the CuO crystals. At around 500 °C, the PVP calcinates as it was determined by differential scanning calorimeter test results, leaving a network of “fibers” similar to the CuO fibers shown in Figure 9 from reference [40]. However, these “fibers” were made of WO_3_ crystals in contact with CuO crystals [41]. This network of metal oxide fibers, or “nanogrid”, exhibits photocatalytic properties.

### 4.3. Nanogrids on Amyloid

While these nanogrids have high efficiency for breaking down benzene and other petroleum hydrocarbons, there will be specific benefits of growing CuWO_4_ from self-assembled amyloid architectures. This process will provide uniformity to particle size, orientation, and spacing between the particles for optimized catalytic effects. Furthermore, amyloids will be acting as CO_2_ sinks, thus providing a completely environmental process for water clean-up. From previous work in the literature [42], it is known that amyloid fibers with alkylamine groups can reversibly bind carbon dioxide through the formation of carbamates. Thermodynamic and kinetic capture-and-release tests show that the rate of carbamate formation is quick enough to capture CO_2_ by dynamic separation in both natural and designed amyloids. The material can also be regenerated by heating it to 100 °C.

### 4.4. TiO_2_ Nanogrids

The effect of amyloid templating on the properties of TiO_2_ nanofibers grown on them is also worth exploring. In our earlier studies [43], the photoresponse of nanofibers annealed at different temperatures was investigated. To achieve this, a photoelectrochemical cell (PEC) was used under a 150 W Xenon lamp (Newport) equipped with an AM1.5 G filter. Figure 10 below illustrates the photocurrent response of the samples upon the on-off illumination vs. time. The data displayed a rapid rise to a steady-state value upon illumination, which is a reproducible effect for several cycles. It can also be observed that the annealing temperature had an effect on the photocurrent.

The highest photocurrent density (around 0.9 mA) was exhibited by the TiO_2_-500 sample, and was a result of the formation of anatase and rutile phases, along with the improved crystallization. Due to this, it could be concluded that nanomats of TiO_2_ annealed at 500 °C have a better photovoltaic performance under visible light. It can also be observed in the graph that there is a decrease in the photocurrent with an increase in temperature. This trend is observed especially in the TiO_2_-700 sample, where the photocurrent decreased dramatically and is possibly due to a larger percentage of the rutile phase (about 50%).

The photocatalytic activity of TiO_2_ is known to depend on grain size, phase, particle morphology, surface and bulk defects, exposed crystalline facets, and specific surface area [44,45]. Figure 10a illustrates TEM images of the TiO_2_ nanogrids, which show the diffraction contrast that was connected to defect structures on the particles. The structures were similar to those observed on anatase nanograins as they transitioned to rutile. Rutile laths were shown to grow on anatase grains [46,47,48]. The results from our experiments illustrated that the best photocatalytic activity under UV and visible light and photocurrent density came from the TiO_2_ sample heat treated at 500 °C, with an anatase to rutile ratio of 90:10 and a particle size of 15–20 nm. This sample can be compared to commercial TiO_2_, which has a ratio of anatase to rutile of 75:25 and a particle size of 24–35 nm.

When comparing our samples with the commercial TiO_2_ P25, there is a stark difference in the phase content and particle size. Previous work from Oak Ridge National Laboratory [49] stated that when the particle size of TiO_2_ was less than 30 nm, a significant increase in the reaction rate of degradation of methylene blue (MB) dye was observed. Other experiments [50,51,52] illustrated that there is an optimum particle size for increased photocatalytic activity and can be attributed to an increased charge carrier recombination rate. This in turn counteracts the increased activity from the increased specific area of TiO_2_ at 600 °C and 700 °C; therefore, the specific surface area is increased with the optimum particle size and could be attributed to the enhanced photocatalytic activity. The increase in surface area would also assist in increasing both the photocatalytic reaction sites and the electron-hole separation efficiency. These in turn promote the absorption of MB dyes on the surface of TiO_2_ nanoparticles. Therefore, the surface and bulk defects, as well as the particle size, are the dominating factors in determining the photochemical properties of pure TiO_2_ mats.

The defect-like contrast observed on the anatase grains is likely to be due to the nucleation and growth of the rutile phase on the surface of the grains. Gouma reported [47] that rutile plates form as fine laths first on the surface of anatase particles via a shear process mechanism. Therefore, any mutualistic effect of the two phases would occur on the anatase grains under transformation. Hybrid processing and polymorphism control can account for novel semiconducting materials with tunable bandgaps and band for use as affordable and highly efficient photo-chemical and photo-electrochemical catalysts activated by visible light.

## 5. Conclusions

Hierarchical processing of functional materials from sol-gel precursors is shown to produce materials with unique or augmented properties. The potential for hybrid organic-inorganic processing is investigated through applications for amyloids and metal oxide nanowires, including skin gas sensing and smart scaffolds. Polymorphism is discussed as an important property of these two types of materials and new avenues of synthesis including amyloids from wheat flour, and hydrothermal for metal oxide nanowires are discussed. Specifics applications, such as amyloids and CA electrospun mats for water remediation, and MoO_3_ nanowires for detecting VOCs in skin gas, are also presented. The use of amyloids to facilitate the growth of metastable metal oxide nanowires is explored. Throughout this perspective, insights into the morphological and structural aspects of material synthesis in correlation with processing and obtained properties are also presented. Many new opportunities using these materials have arisen, leading to a new era of polymorphic organic and inorganic functional systems.

## Figures and Tables

**Figure 1 materials-15-05355-f001:**
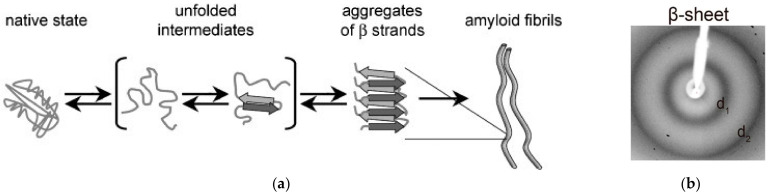
(**a**) Amyloid fibril formation stages [4]; (**b**) XRD pattern typical of β-sheet structures with equatorial (d1) and meridional (d2) reflections (from [8]).

**Figure 2 materials-15-05355-f002:**
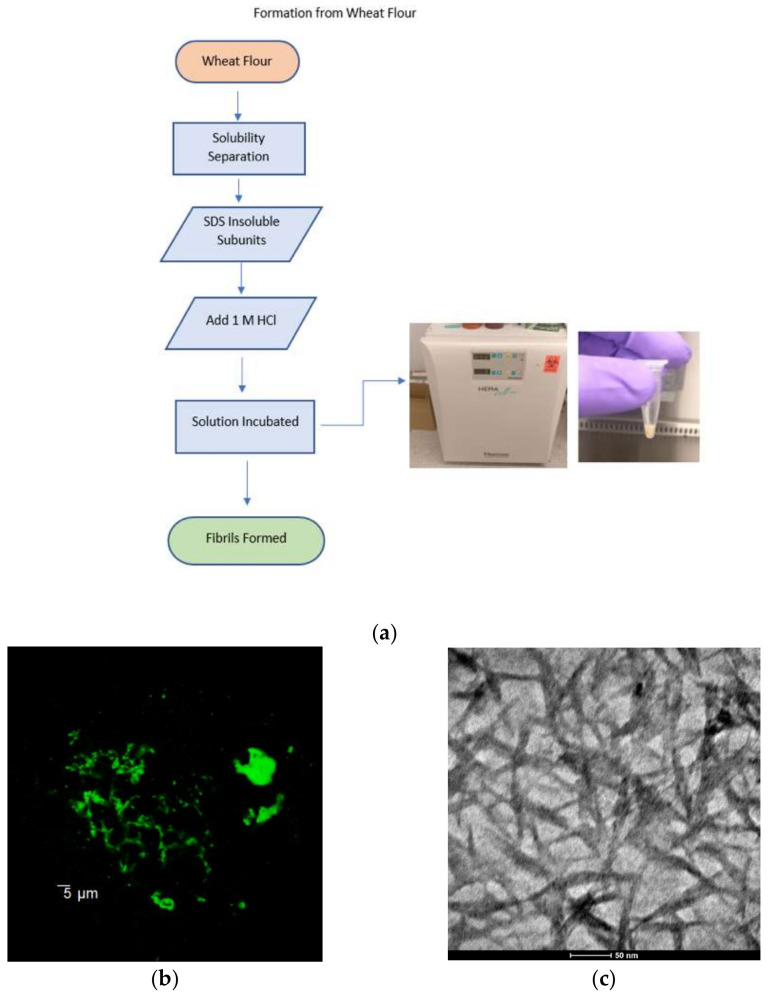
(**a**) Process map of amyloid fibrils from wheat flour; (**b**) confocal image of amyloid fibrils; (**c**) TEM image of amyloid fibrils. Images from Gouma et al. [4].

**Figure 3 materials-15-05355-f003:**
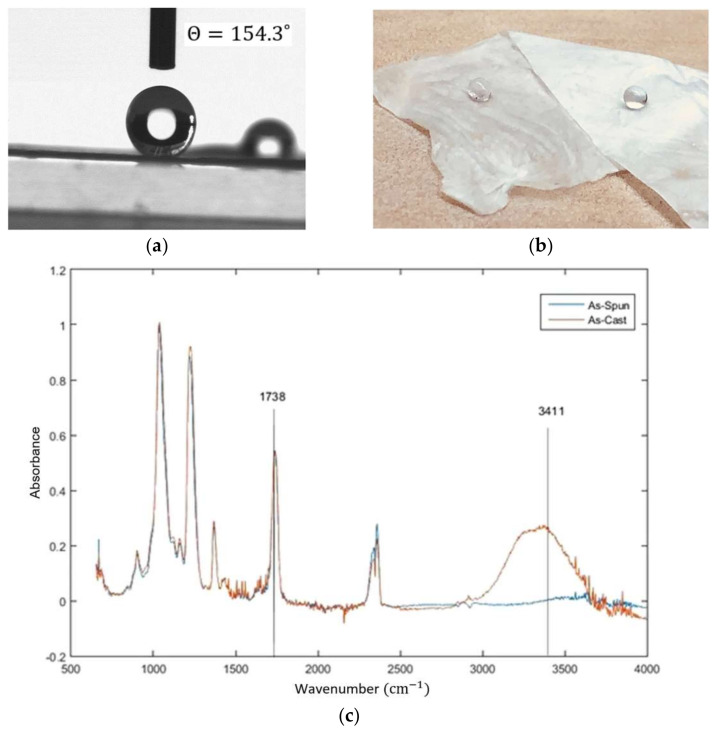
(**a**) Close image of water droplet on super water repellent CA fibrous mats; (**b**) water droplets on super water repellent CA fibrous mats; (**c**) FTIR analysis illustrating the intensity of the hydroxyl band at 3450 cm^−1^ that decreased in the as-spun fiber [25].

**Figure 4 materials-15-05355-f004:**
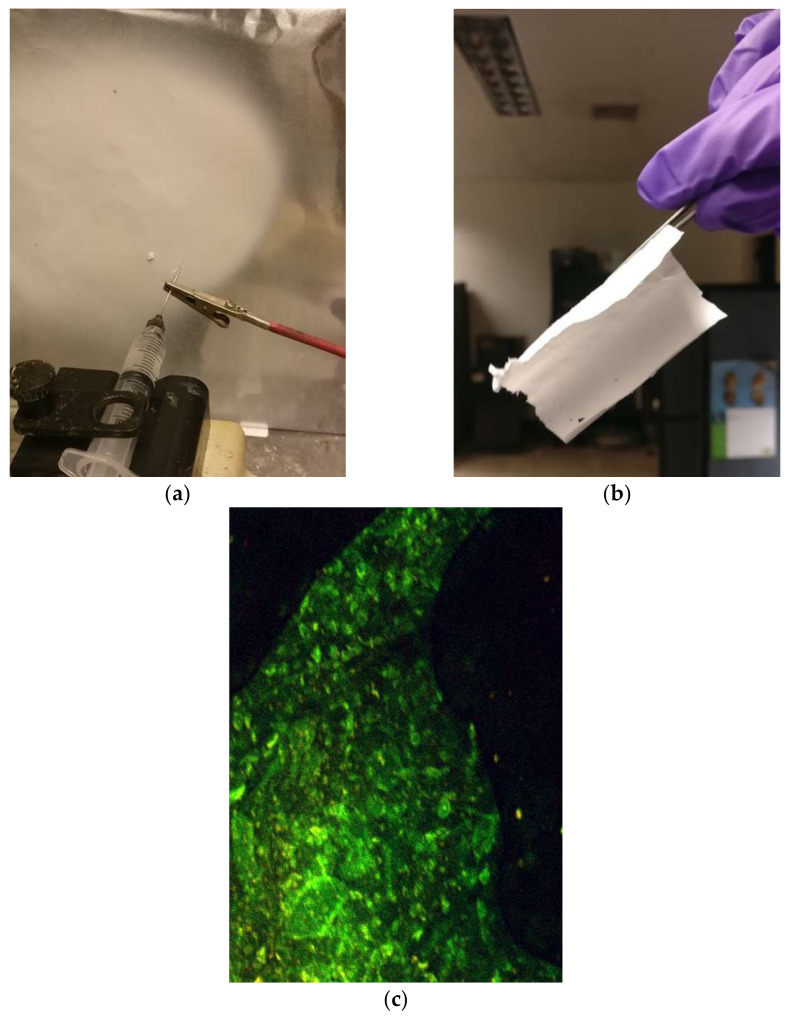
(**a**) Electrospinning of the amyloid and CA solution; (**b**) the electrospun, fibrous mat; (**c**) confocal microscopy of the mat, confirming a uniform dispersion of amyloid fibers in the CA mat.

**Figure 5 materials-15-05355-f005:**
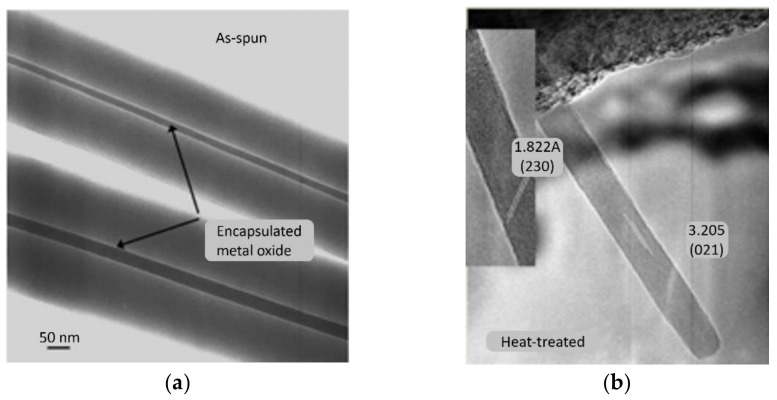
(**a**) TEM images of PVP-MoO_3_ nanocomposite before heat treatment [28]; (**b**) HRTEM image of a MoO_3_ nanowire after heat treatment [28].

**Figure 6 materials-15-05355-f006:**
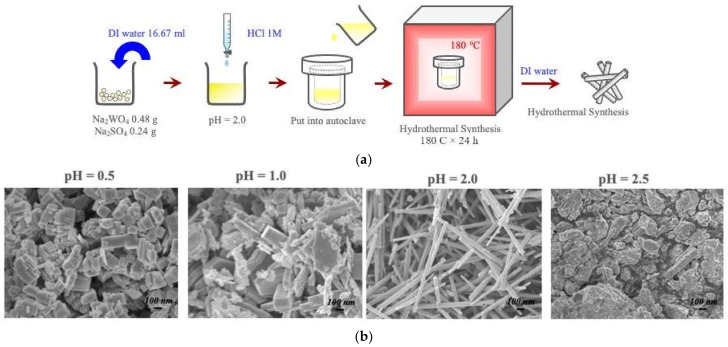
(**a**) Process map for the formation of the metastable hexagonal WO_3_ polymorph via hydrothermal treatment [32]; (**b**) SEM images of the microstructure at varying pH [32]. Nanowires are formed at a pH of 2 with lengths of 10 microns.

**Figure 7 materials-15-05355-f007:**
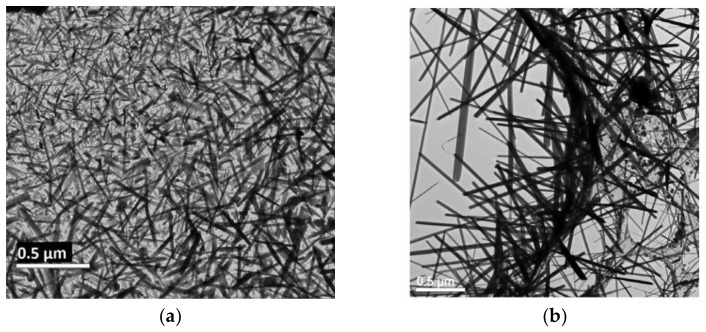
(**a**) TEM image of γ-WO_3_ nanowires grown on silicon nitride grid [33]; (**b**) TEM image of γ-WO_3_ nanowires grown on copper mesh grid [33].

**Figure 8 materials-15-05355-f008:**
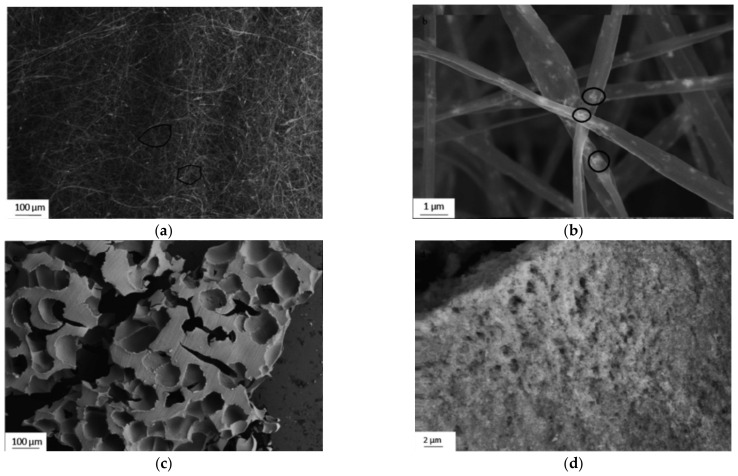
(**a**) SEM images of as-spun fibers, highlighting the formation of a 3D fiber-woven foam (circled) [34]; (**b**) high magnification SEM image of fibers highlighting the sol-phase particles in the fibers (circled) [34]; (**c**) high magnification SEM image of foam surface showing the agglomerated sol particles and nano porosity [34]; (**d**) SEM image of foam structures in the fibers [34].

**Figure 9 materials-15-05355-f009:**
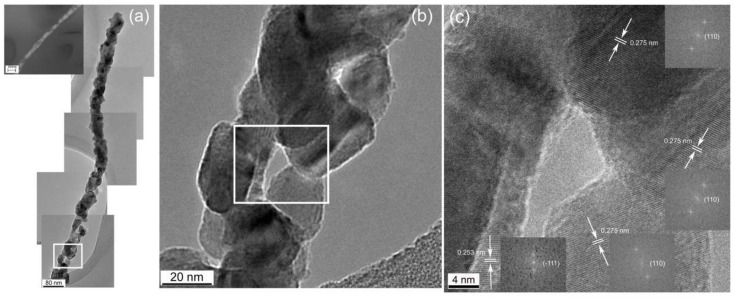
CuO nanowires forming “links in a chain” architecture of nanocrystals with low angle boundaries; this pseudo-monocrystalline porous architecture is called “nanogrids”; (**a**) collage of five HRTEM images of a single chain [40]; (**b**,**c**) englarged views of particle arrangement within the nanowires [40]. The white squares highlight the area captured of the following image.

**Figure 10 materials-15-05355-f010:**
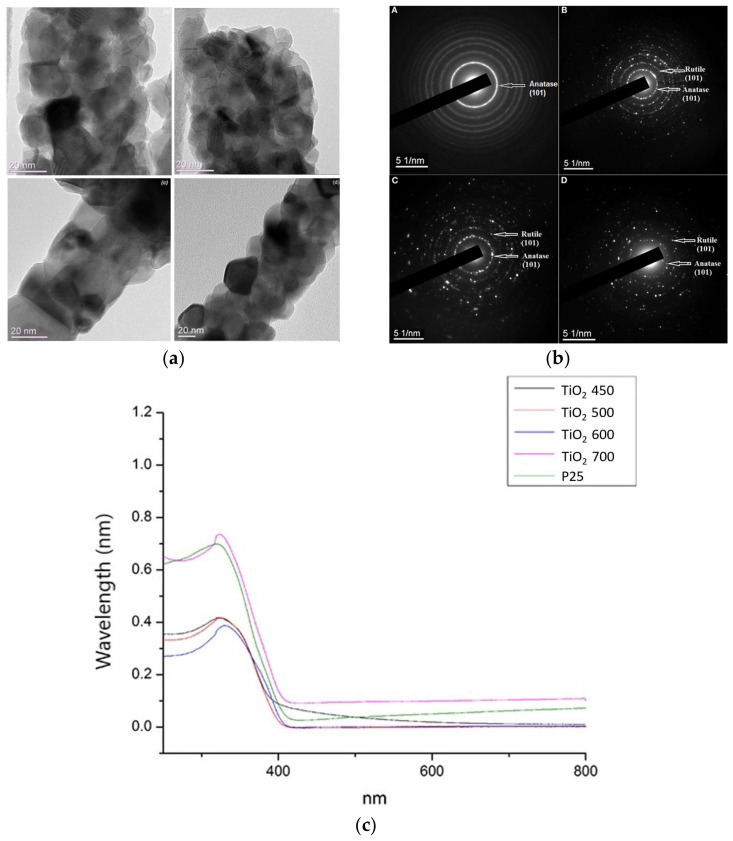
(**a**) TEM images of TiO_2_ nanogrids at 450 °C, 500 °C, 600 °C, and 700 °C; (**b**) SAED patterns of TiO_2_ nanofibers annealed at 450 °C, 500 °C, 600 °C, and 700 °C [43]; (**c**) UV-Vis spectra of P25 and TiO_2_ nanofibers annealed at 450 °C, 500 °C, 600 °C, and 700 °C [43].

## Data Availability

Further information on the data discussed in this manuscript can be found in the reference list below.

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
