# Peer review of "Polymorphic Biological and Inorganic Functional Nanomaterials"

_materials, 2022, doi:10.3390/ma15155355_

Round 1

Reviewer 1 Report

The aim of this MS is to review the latest achievements in the self-assembly and functionality of selected organic and inorganic composite materials. This review is based mostly on the wide original research performed by the authors and supplemented with relevant literature data to support the leading idea. The text is well written and clearly presented, illustrating readers this specific and important topic. Therefore, I gladly recommend this MS for publication.

I find this review complete and certainly relevant, as regarding the intention to cover this specific subject. I have not detected any gap in knowledge, and the cited reference were appropriate (including self-citations).

The MS is clearly written, relevant for the selected topic and very well structured, introducing the reader to novel nanomaterial, current status of the literature knowledge and prospective of each organic (amyloid fibrils) or inorganics (metallic oxides), chosen technique of production and relevant results obtained in characterizing such materials. Finally, merging two subjects, organic and inorganic the authors introduce and describe in detail smart and hybrid scaffolds production and the obtained functionalities leading to the most advanced nanosensors.

I find all citations timely and relevant and no excessive self-citations. All figures/tables/images/schemes are appropriate and clear and easy to interpret/understand in data presenting. The conclusions are concise and consistent with the presented material.

Author Response

To the Reviewer,

Thank you for your kind comments on our submitted manuscript. We did need to make some revision in accordance with the other reviewers' suggestions. Hopefully these changes have only improved your view of the perspective. Please let us know if you believe something was incorrectly revised.

Sincerely,

The Authors

Reviewer 2 Report

The present perspective article submitted by Gouma and Gilmore is a review addressing significant research work on amyloid fibrils, inorganic nanomaterials, and their hybrids. The report consists of three sections – self-assembly of amyloid fibrils, inorganic materials synthesis and properties, and the use of amyloid fibrils in the synthesis of inorganic nanomaterials. In my opinion, the perspective article is suitable for publication in Materials once the authors address the below-mentioned critical issues.

The major issues are

[1] The authors should modify the title – It is too general and does not reflect the content – It could be helpful if authors provide a more appropriate title that directly conveys the article's content. It is unclear what the authors meant by “self-assembly and hierarchical.”

[2] The abstract (vague) and conclusion (general) – The authors should recast them appropriately – in my opinion, the abstract should discuss the critical aspect of the article concisely, and the conclusion should summarize the research area with a note to the future endeavors.

[3] It is unclear what the authors want to convey "We have identified amyloid polymorphism [3] as a key feature controlling the relative affinity of the fibril surfaces" reference [3] is not related to the author's work.

[4] As the authors mentioned, the key aspects of the amyloid fibril are their polymorphic nature as it determines the binding of various organic and inorganic species – This aspect is crucial in the preparation of hybrid materials in combination with inorganic materials – Therefore, it could be helpful for readers if authors pay attention to this aspect while revising the respective sections.

[5] Page 2: correct or modify "condensed state is formed via non-covalent interaction in both cases" both cases mean?

[6] Page 3: The following sentence is confusing “Negative staining obtained from Uranyl Acetate produced adequate contrast and hence the amyloid fibrils appear dark in a bright background”

[7] (a) Page 4: The authors should discuss the reason for the decrease in the –OH stretching frequency in FT-IR (Figure 3c) – (b) The following sentence is vague: “Furthermore, studying the chemistry of the material, and specifically the stretching of the hydroxyl bonds through FTIR was responsible for the change of CA’s hydrophobicity.” (c) The quality of Figure 3c is poor.

[8] Page 9: The authors should recheck the equation 1 “RCH2CH2 R’ + ‘OH RHCH2 R’ + H2O”

[9] Figure 10: The quality of the figure is poor. The Y-axis legend of the figure should be Wavelength (nm) and in Figure TiO2 should be written as TiO2.

[10] References: the journal abbreviations should be in a proper format.

[11] The authors affiliations details are incomplete. 

Author Response

To the Reviewer,

Thank you for your helpful suggestions to the manuscript and suggestion to publish. Please see the list below for the point-by-point response.

[1] The title has been adjusted to better reflect the perspective. ("Polymorphic Biological and Inorganic Functional Nanomaterials")

[2] Please see the restructured abstract and conclusion, which we believe address your comments.

[3] The sentence has been revised to say "In the literature, amyloid polymorphism has been identified [3] as a key feature controlling..." to rectify confusion.

[4] Polymorphism has been emphasized in the revisions to the perspective and a new paragraph explaining the polymorphism and its importance has been added on page 2.

[5] "both cases" has been omitted from the quote as it does not pertain to the perspective.

[6] The sentence has been revised to say "In the confocal images amyloids appear green, however in the TEM images the amyloid fibrils appear dark. This difference is due to the negative staining obtained from Uranyl Acetate." to lessen confusion. 

[7]  a) and b) the last sentence of the paragraph has been revised to better explain the findings and remain within the scope of the perspective.

      c) Figure 3c has been replaced with the same image of a higher quality.

[8] Equation 1 has been revised to correctly match the original article, including the "2" that needed to be subscripted. 

[9] The image has been revised as per comments. 

[10] Journal names and abbreviations have been italicized in the reference list.

[11] Author affiliations have been extended at the top to include Department, City, State, and Country.

Please let us know if you believe something was incorrectly revised.

Sincerely,

The Authors

Reviewer 3 Report

Recommendation: Publish in Materials after minor revisions.

Comments:

The authors presented a perspective on the development of self-assembling nanomaterials with hierarchy. This perspective mainly focuses on amyloid-based and metal oxide nanowire-based derivatives. Some informative discussions have been made on the topic, and therefore, the reviewer recommends its publication in Materials after addressing just a few comments.

1.         Although in this manuscript, both amyloid- and nanowire-based (corresponding to organic- and inorganic-based) hierarchical nanomaterials were discussed, the title of this manuscript is still too broadly defined, as various hierarchical nanomaterials exist yet are not discussed herein. The current title sounds like a title suitable for a book or a comprehensive review in length. Maybe narrow it down to amyloid- and nanowire-based self-assembling hierarchical nanomaterials sounds more specific and could target the appropriate audience.

2.     In figure 8, although appeared in the original publication, it would a better idea for the authors to correct the original publication as it claims the structure in panels (a) and (b) as “honeycomb”. Honeycomb is a well-organized structure with arranged hexagonal (or alike) units with at least short-range order, which is not the case shown in figure 8. It looks like a 3D fiber-woven foam. Please try to find a term or short phrase to better describe it.

3.     If possible, please add a short introduction paragraph describing the progress/challenges in the covered field of research, as well as the motivation/need for the perspective. Starting a sub-class description without a pre-described classification looks less attractive for most audiences, as it may help the audience (who are most likely out-of-field) catch up with current progress and problems encountered in the field. 

Author Response

To the Reviewer,

Thank you for your helpful suggestions to the manuscript and suggestion to publish. Please see the list below for the point-by-point response.

[1] The title has been revised to say "Polymorphic Biological and Inorganic Functional Nanomaterials"

[2] The honeycomb description of the foams has been revised to say 3D fiber-woven foams.

[3] Please see the new introductory paragraph, which we believe addresses your comment.

Please let us know if you believe anything was incorrectly revised.

Sincerely,

The Authors